# Acoustic analogues of three-dimensional topological insulators

Cheng He [1,2,4✉], Hua-Shan Lai[1,4], Bo He[1,4], Si-Yuan Yu[1,2], Xiangyuan Xu[1], Ming-Hui Lu [1,2,3] & Yan-Feng Chen[1,2✉]

Topological insulators (TIs) can host an insulating gapped bulk with conducting gapless boundary states in lower dimensions than the bulk. To date, various kinds of classical wave TIs with gapless symmetry-protected boundary states have been discovered, promising for the efficient confinement and robust guiding of waves. However, for airborne sound, an acoustic analogue of a three-dimensional TI has not been achieved due to its spinless nature. Here, we experimentally demonstrate a three-dimensional topological acoustic crystal with pseudospins using bilayer chiral structures, in which multi-order topological bandgaps are generated step by step via elaborately manipulating the corresponding spatial symmetries. We observe acoustic analogues of 1st-order (two-dimensional gapless surface Dirac cones) and 2nd-order (one-dimensional gapless hinge Dirac dispersion) TIs in three dimensions, supporting robust surface or hinge sound transport. Based solely on spatial symmetry, our work provides a route to engineer the hierarchies of TIs and explore topological devices for three-dimensional spinless systems.

[1] National Laboratory of Solid State Microstructures & Department of Materials Science and Engineering, Nanjing University, Nanjing 210093, China. [2] Collaborative Innovation Center of Advanced Microstructures, Nanjing University, Nanjing 210093, China. [3] Jiangsu Key Laboratory of Artificial Functional Materials, Nanjing University, Nanjing 210093, China. [4]These authors contributed equally: Cheng He, Hua-Shan Lai, Bo He. ✉email: chenghe@nju.edu.cn; yfchen@nju.edu.cn

Topology refers to the conserved properties of spaces under continuous deformations. The bandstructures of crystals in energy-momentum space can also possess certain topological behaviours. One hallmark of topological phases is the existence of gapless boundary states at the interface between two topologically distinct materials, leading to robust wave transport in the presence of disorders, defects or bends, which is highly desired in many practical scenarios. In the past decades, various kinds of topological phases of matter have been discovered in condensed-matter physics[1,2] and soon thereafter were extended to photonic[3–7] and phononic[8–12] crystals due to the similar bandstructures of photons and phonons. In particular, photonic and phononic topological insulators (TIs) can support gapless boundary states in lower dimensions than the bulk, leading to robust wave transport in the presence of disorders, defects or bends, which is highly desirable in many practical scenarios[13–20]. In two-dimensional (2D) systems, photonic TIs have shown great application potential in the realization of robust optical delay lines[13], topologically protected waveguides[14,15], topological quantum emitters[16] and topological insulator lasers[17]. Moreover, three dimensional (3D) and even higher dimensional photonic TIs can be realized, benefiting from the two intrinsic polarizations and controllable coupling between the electric and magnetic components of photons[21–23] or synthetic dimensions[24–26]. For spinless airborne sound, however, the lack of a spin degree of freedom and a magnetic response make it extremely difficult, if not impossible, to directly mimic the spin and spin-orbit coupling of solid-state TIs. Although some 2D acoustic TIs with one dimensional (1D) topologically protected edge channels have been considered to artificially construct acoustic pseudospins based on in-plane spatial symmetries[18,19], the topology of the extra out-of-plane direction is difficult to achieve the surface Dirac cone that is linear dispersion along an arbitrary direction[27,28]. To our knowledge, it is still challenging to achieve an acoustic analogue of 3D TI exhibiting 2D massless Dirac quasiparticles near the surface Dirac cone for a spinless system.

Recently, topological Weyl systems have been implemented in 3D acoustic crystals showing unique helicoidal surface dispersion and one-way negative refraction for the lateral surfaces[29–32]. Compared to 3D TIs with gapless surface Dirac cones in full bulk bandgaps, the bulk bandstructures of these Weyl systems contain paired linearly touching points (Weyl points) with opposite Berry charges. Due to the lack of full bulk bandgaps, the sound could be scattered into crystals when meeting defects or bends, which inevitably jeopardize the efficiency of sound transport. Furthermore, in addition to the existence of the 2D (3–1 dimension) topological surface states in 3D acoustic TIs, 3D systems can also support 1D (3–2 dimension) hinge states and zero-dimensional (0D, 3–3 dimension) corner states—the so-called hierarchies of a higher-order TI, owing to the bulk-boundary correspondence[33]. Although topologically gapped 0D corner states have been demonstrated in 2D[34–36] and even in 3D[37,38] acoustic crystals, they reside inside the midgap (not gapless in energy-momentum space) that cannot be used to guide and route sound. To date, a 3D 2nd-order acoustic TI with gapless hinge states exhibiting topological 1D hinge sound transport in 3D geometries remains unexplored.

To realize an acoustic analogue of 3D TI, there are three crucial criteria that must be simultaneously satisfied: (i) a full 3D acoustic bulk bandgap (confining the sound in all three directions) should open by breaking the spatial symmetry; (ii) a pair of acoustic pseudospins (satisfying Kramers degeneracy) should be constructed by increasing the spatial symmetry; and (iii) a band inversion should occur (making it topological) by manipulating the interlayer and intralayer couplings. In this paper, we demonstrate the realization of a multi-order acoustic analogue of 3D TI

with simultaneous fulfillment of all the above requirements by introducing a 3D acoustic crystal with bilayer chiral structure to precisely break the specific spatial symmetry in a step by step manner. The topological transition associated with both the acoustic analogues of 3D 1st- and 2nd-order TIs are clearly shown in this work.

## Results

**From Weyl points to 3D double Dirac cone**. A general approach to realize a 3D TI can begin with 3D Dirac cones that are linear dispersions of the bulk bandstructures along all three directions. As illustrated in Fig. 1a, we can lift the 3D bulk Dirac cone by introducing interactions, such as spin-orbit coupling, to create a 1st-order topologically full bandgap wherein the 2D surface Dirac cone can be achieved with massless Dirac quasiparticles at the surfaces. We may further lift such a 2D surface Dirac cone by breaking either the spatial or time-reversal symmetry, to create a 1D topological edge of the 2D surface, which is a 3D 2nd-order TI with 1D gapless hinge states. However, for a spinless acoustic system, obtaining the 3D acoustic bulk Dirac cone (a 4-fold degeneracy is required when considering the spin degree of freedom) is not straightforward due to the lack of two intrinsic spins or polarizations[39]. Inspired by recently the discovered Weyl acoustic systems with chiral structures[29,30,32,40,41], we elaborately control the parameters to construct iso-frequency Weyl points with opposite topological charges (Supplementary Fig. 1). As shown in Fig. 1b, the 3D acoustic crystal has a honeycomb lattice with six spiral air channels connecting the top and bottom planes in each unit cell. In contrast to the conventional approach from Dirac to Weyl systems by breaking symmetries, here we increase the spatial symmetry by doubling the lattice along the $z$-axis[42]. The Weyl point at the $H_1$ and $H_1'$ points can be folded into K and K' to form two 3D double Dirac cones, attributed to the Brillouin zone folding mechanism (Fig. 1c). Then, we can break this half-lattice spatial translation symmetry by enlarging the lower spiral channels to create a topologically full bandgap (Fig. 1d). The corresponding bulk bandstructures are shown in Fig. 1e–g. Consequently, we obtain an acoustic analogue of 3D 1st-order TI, which can host 2D gapless surface Dirac cones.

**Acoustic analogue of 3D 1st-order TI**. Our samples are fabricated by 3D printing with photopolymer materials, which are opposite structures of the air channels shown in Fig. 1d. In the experiment, we use four uniform rhomb–prisms of 3D acoustic crystals as segments to create a 2D topological domain wall in the $yz$ plane (reversed $z$ direction with opposite sign of effective mass)[43], and to check the robustness of sound transport in the presence of bends or a splitter, as shown in Fig. 2a. Each rhomb–prism has $7 \times 7 \times 7$ unit cells with full dimensions of $21 \times 21 \times 28.3$ cm (Supplementary Fig. 2). The numerically computed projected bandstructures are shown in Fig. 2b and host gapless surface Dirac cones in a full bandgap. A magnified 3D view of the surface states near the degenerate point (Fig. 2c) clearly shows conical-like dispersion. To verify such gapless surface states, we experimentally measure the transmission spectra along the domain wall and through the bulk, as shown in Fig. 2d. We observe more than a 30-dB enhancement of the surface transmission from ~5.4–6.1 kHz, with a 10% relative topological bandgap width. To verify the robust sound transport and acoustic pseudospin-momentum locking, we assemble the above four segments to form a 60° bend, a 120° bend, and a splitter configuration. The experimental results in Fig. 2e show that sharp bends do not decrease the surface transmission, and a particular acoustic pseudospin is well locked to its momentum which cannot propagate through the cross port by a flipping pseudospin[18].

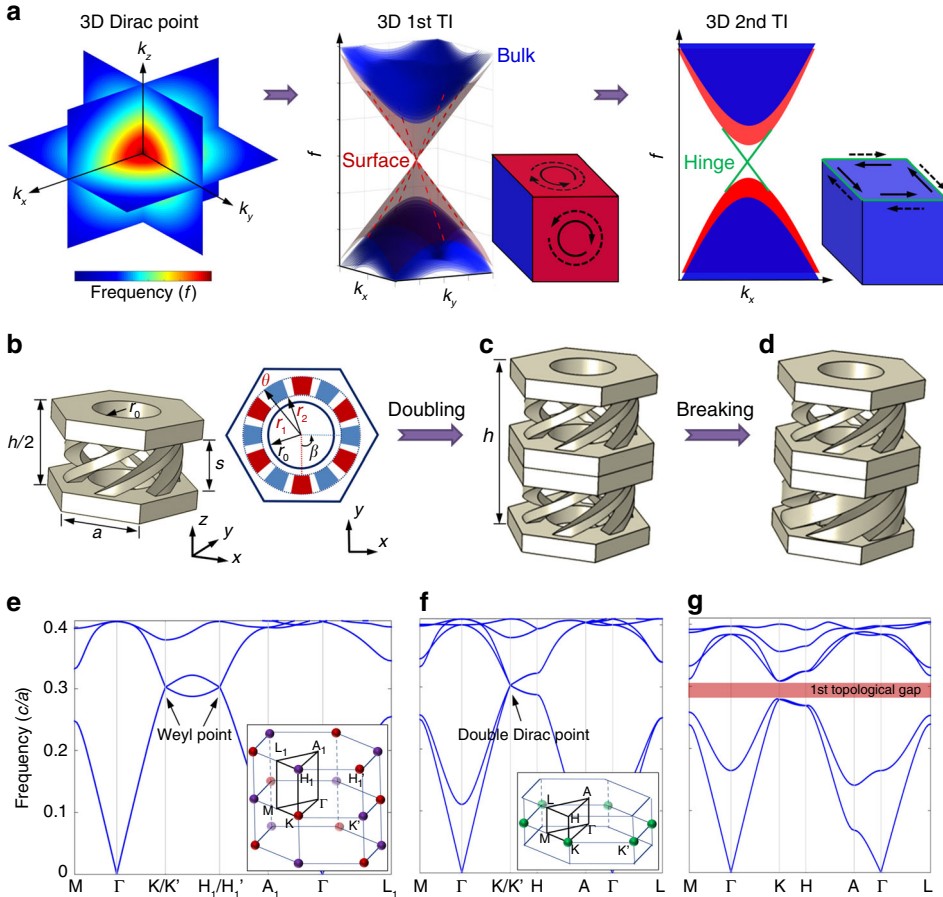

**Fig. 1 From Dirac cones to 3D 1st- and 2nd-order acoustic analogues of TIs. a** Left side: visualization of a 3D Dirac cone for bulk bands, where the colour scale represents frequency. Middle: 1st-order TI case by splitting the bulk Dirac degeneracy, where the red planes show the 2D topological surface states. Right side: 2nd-order TI case after further splitting the surface degeneracy, where the green lines show the 1D gapless hinge states. **b** Single-layer unit cell and top-down view of the honeycomb lattice with the chiral structures wherein sound propagates. The structure parameters are $h = 7/a$, $s = 2/a$, and $r_0 = 1.3/a$, where $a = \sqrt{3}$ cm. The parameters of the spiral channels are $r_1 = \sqrt{3}a/2 - 0.2$ cm, $r_2 = r_0 + 0.2$ cm and $\theta = 15°$ with spiral angle $\beta = 90°$. The colours blue and red represent the bottom and top channels, respectively. **c** Bilayer unit cell case. **d** Breaking the $h/2$ spatial translation symmetry by enlarging the spiral channels of the lower layer to $r_1 = \sqrt{3}a/2 - 0.05$ cm, $r_2 = r_0 + 0.05$ cm and $\theta = 40°$. **e–g** The bulk bandstructures corresponding to the unite cells in panels **b–d**, respectively. The insets show the Brillouin zones, where the red and purple dots denote the Weyl points with opposite charges and the green dots denote the 3D double degenerate Dirac points.

In addition, 2D slices of simulated acoustic pressure distributions are given in Fig. 2f (Supplementary Figs. 3–4). To further reveal the acoustic pseudospin-momentum locking, we numerically analyse the pseudospin textures (Supplementary Fig. 5). The Acoustic pseudospins here are composed of in-phase and out-of-phase vibrations in two different layers[44].

Furthermore, our acoustic analogue of 3D TI can support 2D gapless surface Dirac cones in the $xy$ plane with a hard or open boundary condition instead of a domain wall, which is highly desired in applications. We place a plastic board on both the top and bottom surfaces to act as hard boundaries. The 2D gapless surface Dirac cones still exist (the numerically calculated projected bandstructures in the $xy$ plane shown in Fig. 3a). The corresponding 3D view of the simulated surface Dirac cone is shown in Fig. 3b. To experimentally map out the surface Dirac cones of the 3D acoustic model, we scan the acoustic pressure fields of the top surface at various frequencies. Thus, at each frequency, the Bloch momentum of the $k_{xy}$ plane can be obtained by a Fourier transformation of the field distributions in real space. The measured surface Dirac cone at various frequency slices is plotted in Fig. 3c, which matches well with the numerical results

in Fig. 3b. The surface Dirac point appears at ~5.8 kHz. Interestingly, in this case, topological behaviours only occur at the top surface of the 3D acoustic crystals due to the chiral structures, which are also confirmed by our experiments shown in Fig. 3d and the simulated field distribution shown in Fig. 3e. If we use an open or a soft boundary condition, the surface Dirac cones will appear at the bottom surface (Supplementary Figs. 6–9).

It should be noted that there are two gapless surface Dirac cones near the projection of K and K′ (K′ is not shown) in the whole surface Brillouin zone, which can be treated as an acoustic analogue of a 3D weak TI[45]. In our model, the gapless surface Dirac cones can be found in both (100) surface ($yz$-domain wall) and (001) surface ($xy$ plane), while the surface states are gapped in (010) surface ($xz$ domain wall) since the K and K′ valleys are projected onto the same point (Supplementary Fig. 10). The bulk topology of our model can be verified via the Wilson-loop approach[46,47]. Although the weak TI is not as robust as strong TI (in terms of the odd number of surface Dirac cones), the topologically protected sound transport of our 3D acoustic crystal is largely maintained. We also numerically study the influence of surface condition on gapless topological states in various surface

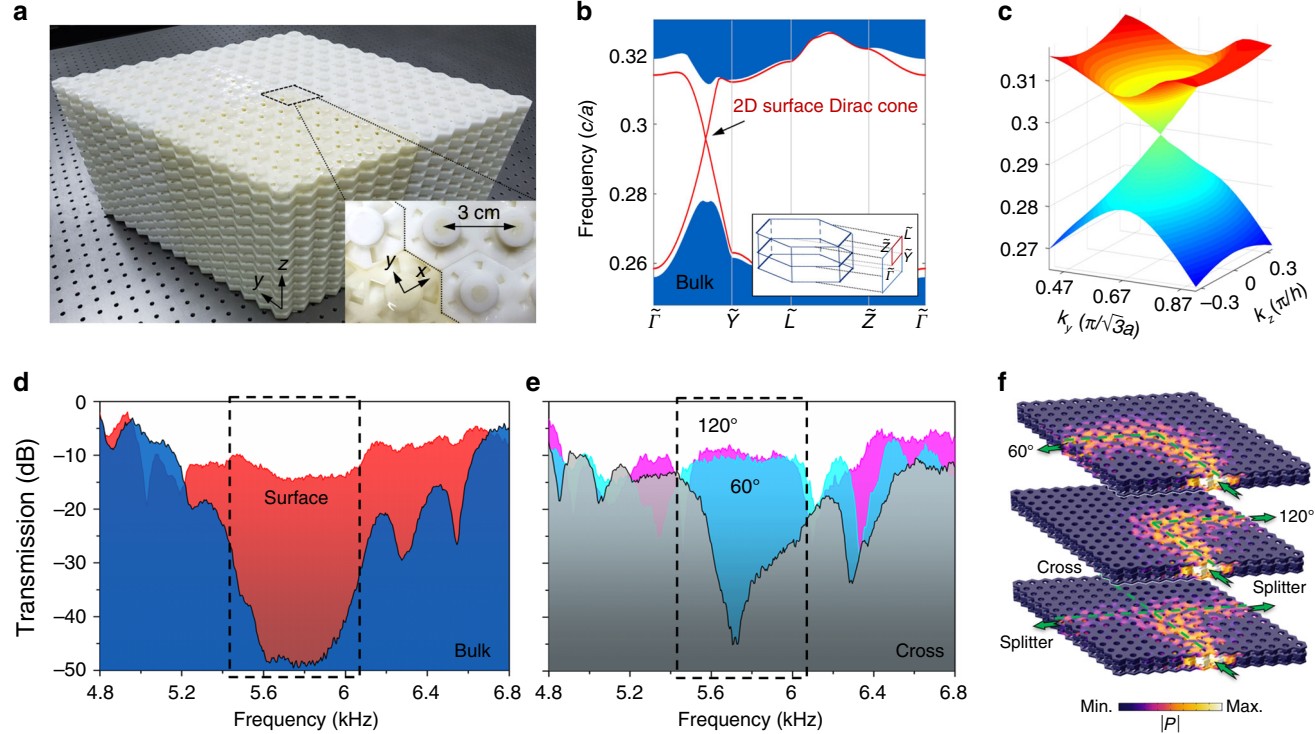

**Fig. 2 Topological surface on *yz*-domain wall. a** Picture of the experimental sample constructed by four rhomb–prisms. Each segment has 7 × 7 × 7 unit cells (e.g. a yellow rhomb–prism). The inset shows a top-down view of the zigzag interface. **b** Projected bandstructures in the $k_{yz}$ plane, where the red lines represent the gapless Dirac-like surface states. The inset shows the highly symmetric directions of the surface Brillouin zone. **c** 3D view of the simulated surface Dirac cone. **d** Measured transmission spectra for the domain wall (red) and the bulk (blue) along the $\tilde{\Gamma}\tilde{Y}$ direction. The dashed box indicates the frequency range of the bulk bandgap. **e** Measured transmission spectra for a 60° bend (cyan), a 120° bend (magenta), and a splitter configuration (grey). **f** Slices of the simulated acoustic pressure distribution at a frequency of 5.8 kHz, where the colour scale represents the absolute value of acoustic pressure (P).

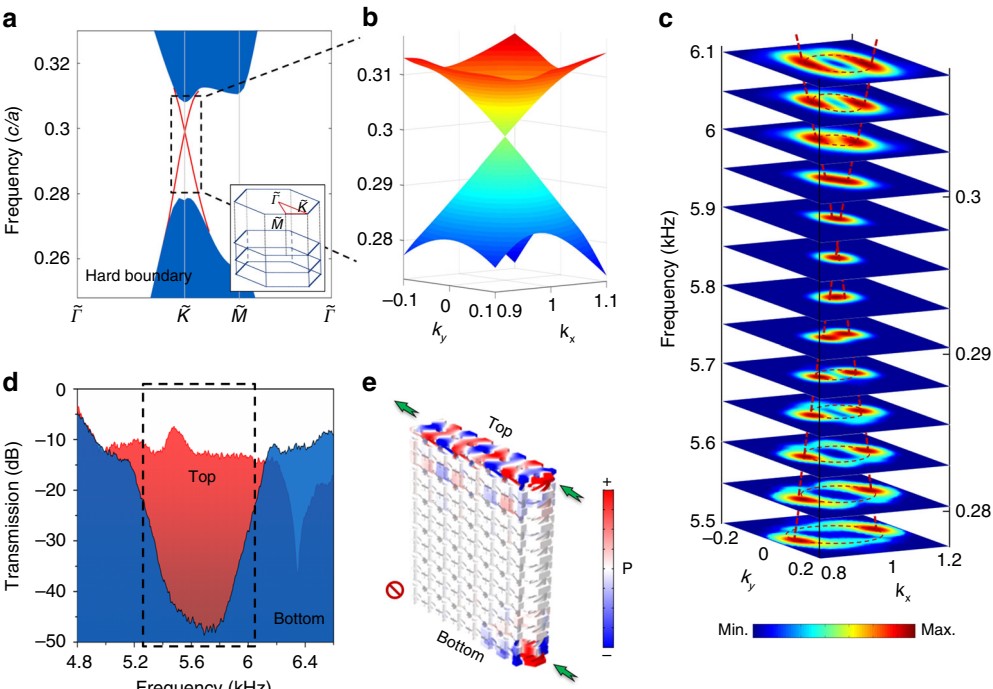

**Fig. 3 Topological surface in the *xy* plane. a** Projected bandstructures in the $k_{xy}$ plane with a hard boundary condition, where the red lines represent surface states. The inset shows the surface Brillouin zone. **b** 3D view of the simulated surface Dirac cone. Here, $k_x$ and $k_y$ have units of $4\pi/9\,\text{cm}^{-1}$. **c** Experimentally measured iso-frequency slices with Bloch momenta for the surface states. The colour scale represents the acoustic energy density. The dashed lines are drawn to guide the eye. **d** Measured transmission spectra for the top (red) and bottom (blue) surfaces along the $\tilde{\Gamma}\tilde{K}$ direction, where the colour scale represents the energy density. **e** A slice of the simulated acoustic pressure distribution with acoustic source excited at both top and bottom surfaces at a frequency of 5.8 kHz, where the colour scale represents the acoustic pressure.

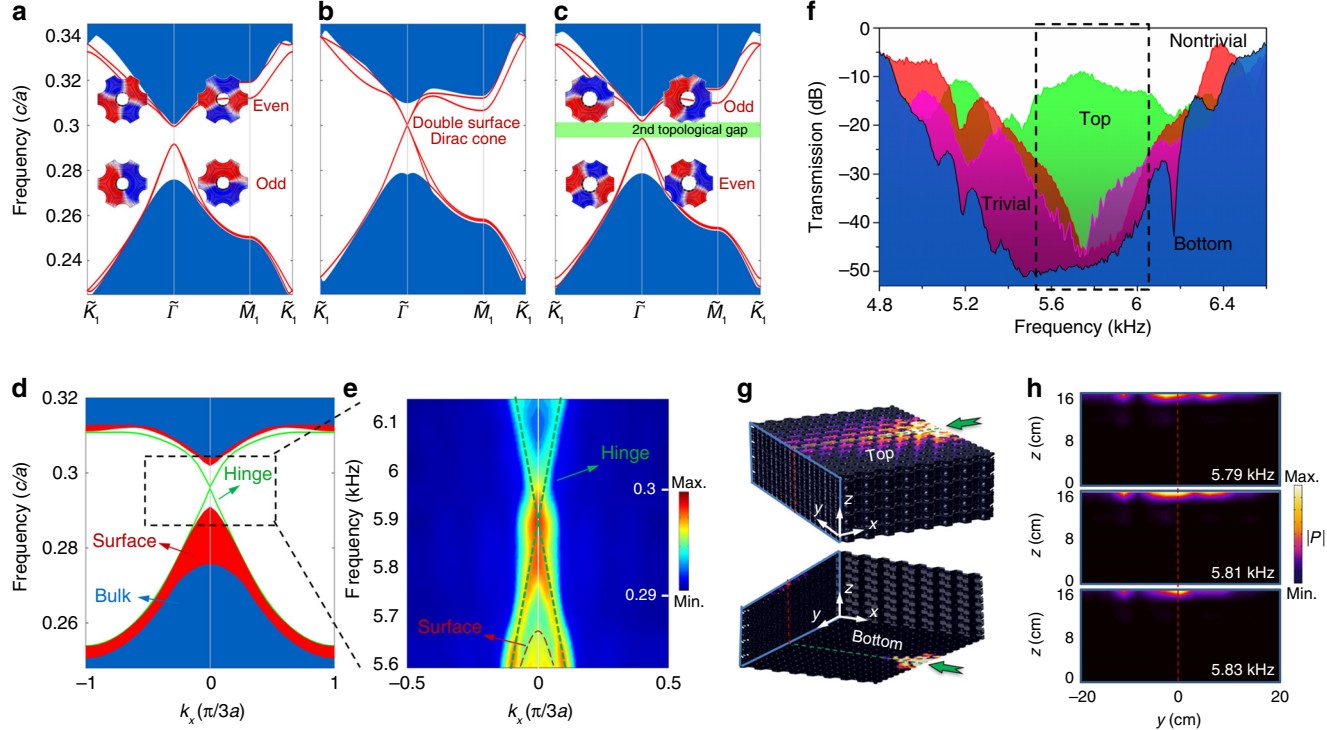

**Fig. 4 Observation of the acoustic analogue of 3D 2nd-order TI. a** Trivial surface bandgap when increasing the radii of the side holes. **b** Double-surface Dirac cone case. **c** Nontrivial surface bandgap when increasing the radius of the center hole. The insets show a top-down view of the acoustic fields of the surface states. **d** Numerically calculated projection of the 3D bandstructures onto the 1D $k_x$ direction, where the colours blue, red and green represent the bulk, surface and hinge states, respectively. **e** Measured dispersion of hinge states, where the colour scale represents the acoustic energy density. The dashed lines are drawn to guide the eye. **f** Experimental transmission spectra of the trivial surface case (magenta), nontrivial surface case (red), bottom (blue) and top (green) hinges. The dashed box indicates the frequency range for the surface bandgap. **g** Simulated acoustic pressure distributions for the top and bottom hinges at a frequency of 5.81 kHz. **h** Measured acoustic pressure distributions in the outgoing plane at various frequencies. The colour scale represents the absolute value of acoustic pressure.

configurations. However, if we use alternate chiral channels to obtain full bulk gap, only trivial topological states can be found (Supplementary Figs. 11–19).

**Acoustic analogue of 3D 2nd-order TI.** More importantly, we can further utilize such two 2D surface Dirac cones to construct an acoustic analogue of 3D 2nd-order acoustic analogue of TI with 1D gapless hinge states. As shown in Fig. 4a–c, based on Brillouin zone folding in the $xy$ plane, we fold these two cones to form a double-surface Dirac cone considering a larger unit cell, which is three times larger than the original one (Supplementary Fig. 20). By selectively increasing the radii of either the center hole or the side holes (from $1.3/a$ to $1.6/a$), we break the 2D gapless surface states to a full bandgap associated with a band inversion for the surface states[48]. The odd nature of the lower two surface states and even nature of the upper two surface states for the trivial case (Fig. 4a) are reversed to create a 2nd-order topological bandgap (Fig. 4c). The numerical 1D $k_x$-direction projected bandstructures of the 3D system (Fig. 4d) reveal that the gapless 1D hinge Dirac dispersion can be achieved. Our experimental measurement matches well with the numerical result as shown in Fig. 4e, where the hinge Dirac point appears near the frequency of 5.9 kHz (experimental sample is shown in Supplementary Fig. 21). We also experimentally measured the transmission spectra of trivial and nontrivial 2D surface states with a surface bandgap from ~5.6–6 kHz, but the transmission for the 1D hinge states of the top surface maintains a high value, as shown in Fig. 4f. Figure 4g is a simulated result that shows this unique propagation. To verify such hinge states that are confined

along the 1D direction, we experimentally scan the acoustic pressure fields of the outgoing plane at various frequencies; the results clearly show that the sound is confined and guided along the 1D hinge (Fig. 4h). The 1D hinge states can also be observed along the $y$ direction or under soft boundary condition (Supplementary Figs. 22–23). Moreover, such gapless hinge states can be further lifted to support higher-order topological states deemed as 3rd-order corner states in 3D topological acoustic crystals.

**Discussion**

It should be noticed that the pseudospins of our acoustic systems are based on spatial symmetries, which are unlike the electrons with two intrinsic spins naturally satisfying the Kramer degeneracy[44]. Thus, besides bulk topology (Supplementary Figs. 24–28), the boundary condition also plays a key role to realize the gapless surface or hinge states. In our case, the top surface (hinge) of the current model can support suitable acoustic pseudospin states while the bottom surface (hinge) does not support these states for hard boundary condition, and vice versa for soft boundary condition[49]. This is the so-called boundary-obstructed topological phase[46] or fragile TI[50–52]. On the other hand, although such fragile topological phase suffers from some limitations, it provides us an opportunity to develop on-demand gapless acoustic surface Dirac cones by choosing a proper surface condition.

To conclude, we have experimentally realized acoustic analogues of 3D 1st- and 2nd-order TIs, where robust 2D surface transport and 1D hinge transport are observed in the bandgaps. Our work extends the scope of acoustic analogues of TIs from 2D

to 3D, and opens up opportunities for further exploring acoustic topological phases based on 2D topological surfaces. The ability to design an analogue of 3D TI in a spinless system based solely on spatial symmetry deepens our understanding of the role of spins and the dimensional hierarchies in topological physics. Compared to a recently realized 3D photonic weak TI benefiting from electromagnetic coupling[23], such acoustic pseudospins and high-order acoustic topological states may provide a new possibility to create an artificially magnetic-like response for sound. Our systems can also serve as a table-top platform to explore unique acoustic devices, such as robust acoustic imaging based on 2D topological surfaces, and robust sensing based on 1D topological hinges in 3D geometries.

## Methods

**Experiments**. In experiments, the samples are fabricated by 3D printing with photopolymer materials. Due to the fabrication tolerance (~±0.1 mm, relative tolerance less than 1%) of different samples, the frequency ranges show a slight red or blue shift relative to each other. In experiments, we use a commercial loudspeaker (AMT-47) to generate a white noise as acoustic sources. A detector (a small-size microphone) is inserted in the sample to detect the transmission spectra and scan the pressure fields. The data are analysed with NI cDAQ-9185 (NI 9250 & NI9260). The experimental transmission spectra in the main text (Figs. 2d, e, 3d, and 4f) are normalized by the acoustic wave transmission through the same distance in air.

**Simulations**. Bandstructures and field distributions in the main text (Figs. 1e–g, 2b, c, f, 3a–b, e, 4a–d, and g) are numerically calculated by the commercial finite-element-method software (COMSOL MULTIPHYSICS). The numerical models are constructed using only acoustic cavities with connecting chiral air tubes due to the large acoustic impedance mismatch between air and the polymer materials, which can be treat as hard boundaries. The density and velocity of sound are chosen to be 1.25 kg m$^{-3}$ and 343 m s$^{-1}$, respectively. The numerical results match well with our experimental measurements.

## Data availability

All the data supporting the findings in this study are available in the manuscript and in the Supplementary Information. Further data and methods are available from the corresponding authors upon request.

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

## Acknowledgements

The work was jointly supported by the National Key R&D Program of China (grant Nos. 2017YFA0305100 and 2017YFA0303702) and the National Natural Science Foundation of China (grant Nos. 11874196, 11890700, 11625418, 51732006, 51721001 and 51702152). We also acknowledge the support of the Fundamental Research Funds for the Central Universities (grant No. 021314380097).

## Author contributions

C.H. conceived the idea. C.H. and H.S.L. contributed to the design of the structures and performed the theoretical modeling. C.H. and B.H. carried out the measurements with the help of S.Y.Y. and X.X. C.H., M.H.L. and Y.F.C. contributed to the discussion of the results. C.H. and Y.F.C. supervised all the aspects of the work and managed the project.

## Competing interests

The authors declare no competing interests.
