## [Peer Review File · Nature Communications]

Reviewers' comments:

Reviewer #1 (Remarks to the Author):

In this work, authors claimed that they realized 3D acoustic topological insulator. The experimental results are clear, but authors incompletely proved what they claimed.

It is well-known (authors also mentioned at start) that bulk-boundary correspondence lies at the foundation of topological physics. Authors demonstrated surface/hinge states, but throughout the whole paper, nothing has been shown on their corresponding bulk topology.

Authors stated that two topologically distinct materials can give rise to an interface which could host gapless boundary states. In Fig. 3, they only observed topological surface state between the top surface of acoustic crystal and a hard plate. What is the topological distinction between the acoustic crystal and the hard plate?

The same crystal standing on a lower hard plate gives rise to another boundary which however cannot support surface state. How to explain that with the bulk-boundary correspondence theorem?

What's the topological invariant of 2nd-order topological insulator?

Reviewer #2 (Remarks to the Author):

This paper studies a 3D acoustic structure that simulates a 3D topological insulator exhibiting surface Dirac cones and a second order topological insulator exhibiting 1D gapless hinge states. The design and experiment are both new. I particularly like the idea of folding two Weyl points to form a 3D Dirac point.

My comments are as follows.

1. The observed Dirac cone in Fig. 3, from my point of view, can probably arise from the honeycomb structure at the top surface (just like graphene), and thus may not necessarily link to the bulk topology. Moreover, I don't understand why the bottom surface has no surface states. I hope the authors can clarify these points more clearly.

2. In Fig. 2, I guess all surface states are excited in the measurement. So the real situation is more complicated than the simulation in Fig. 2f. For example, what is the spin-momentum locking in this situation?

Reviewer #3 (Remarks to the Author):

The work looks interesting. Nevertheless, clarification of some aspects could improve the presentation. In particular:

1. Abstract, lines 52-60: it would be helpful if the authors elaborate more on the differences between the current work and Refs. 29-32.

2. Line 54: what do the authors mean by "whereas sound could be scattered into crystals due to the gapless bulk bandstructures that inevitably jeopardize the efficiency of sound transport."? The bulk bandstructures shown in Refs. 28-32 look gapped.

3. Figure 1: subplot (a), the 3D Dirac point and the dispersion of the surface states looks quite similar to subplots (a) and (c) in Figure 1 in Ref. 28. For the surface states, what is different in current work in the ability to guide sound along the surface (which was impossible in Ref. 28)? Is the main novelty

here are the hinge states?

4. Figures 2-3: it appears that the presented guided surface waves are a result of numerical simulations. Are there any measurements of such waves conducted in the experimental system?

5. Figure 4: it would be helpful if the authors explain more about the differences between (d)-top and (d)-bottom.

Reviewers' comments:

Comment of Reviewer #1:

In this work, authors claimed that they realized 3D acoustic topological insulator. The experimental results are clear, but authors incompletely proved what they claimed.

Our response: We do appreciate the reviewer for his/her positive comments of our experiments. The main concern is about how to explain the topology of our model.

Comment of Reviewer #1:

It is well-known (authors also mentioned at start) that bulk-boundary correspondence lies at the foundation of topological physics. Authors demonstrated surface/hinge states, but throughout the whole paper, nothing has been shown on their corresponding bulk topology.

Our response: We thank the reviewer for his/her comments. We agree that the bulk-boundary correspondence lies at the foundation of topological physics. Here, we use the Wilson-loop approach to explain the topology of our bulk bands.

To match the surface Brillouin zones of Fig 2b in the main text and Fig. S10c in the supplementary materials, we use an orthogonal unit cell to carry out our calculations (Fig. R1a). The orthogonal unit cell is two times larger than the primary unit cell (Fig. 1d in the main text). Therefore, the first two bulk bands below the bandgap (Fig. 1g in the main text) are folded into four bands; however, their topological properties are the same.

For the topology on the yz surface (Fig. 2 in the main text), we numerically calculate the Wilson loop of all four bulk bands below the bandgap along the x direction. As shown in Fig. R1b, their Berry phases interpolate across the whole $-\pi$ to $+\pi$ area with a gapless feature, showing its topological nontrivial character similar to Z_2 topological insulator. On the contrary, for the xz surface (Fig. S10 in the supplementary materials), the Berry phases, with a Wilson loop along the y direction, cannot interpolate across the whole $-\pi$ to $+\pi$ area, indicating the trivial behavior of the xz surface (Fig. R1c).

Fig. R1 | Hybrid Wannier centres. **a**, Schematic of an orthogonal unit cell. **b**, Eigenvalues of the Wilson loop of four bulk bands below the bandgap along the x direction corresponding to Fig. 2 in the main text. **c**, Eigenvalues of the Wilson loop along the y direction corresponding to Fig. S10 in the supplementary materials.

Action: We added above figures into our supplementary materials (Fig. S24).

Comment of Reviewer #1:

Authors stated that two topologically distinct materials can give rise to an interface which could host

gapless boundary states. In Fig.3, they only observed topological surface state between the top surface of acoustic crystal and a hard plate. What is the topological distinction between the acoustic crystal and the hard plate?

Our response: We thank the reviewer for his/her comments. Our acoustic crystal is topologically nontrivial while the hard/soft plate is topologically trivial. To further explain the difference of the top and bottom surfaces, we numerically studied two kinds of two-layer configurations: BA-stacking and AB-stacking configurations, where A represents the small spiral channel layer while B represents the large spiral channel layer. As shown in Fig. R2, the surface states prefer A surface layer under hard boundary condition, while they prefer B surface layer under soft boundary condition. This situation is similar to a 3D topological photonic case with perfect-electric-conductor boundary or perfect-magnetic-conductor boundary. [Fig. 5 of Ref: T. Ochiai, Gapless surface states originating from accidentally degenerate quadratic band touching, *Phys. Rev. A* **96**, 043842 (2017)].

Fig. R2 | Intact boundary cases with hard and soft boundaries. **a**, BA-stacking supercell configuration, where A represents the small spiral channel layer while B represents the large spiral channel layer. **b**, Projected bandstructures and Bloch field distributions with hard boundaries, where the surface states located at the top (A layer). **c**, Soft boundary case, where the surface states located at the bottom (B layer). **d**, AB-stacking supercell configuration. **e**, Hard boundary case, where the surface states located at the bottom (A layer). **f**, Soft boundary case, where the surface states located at the top (B layer).

Action: We added above figure and reference in our supplementary materials (Fig. S15).

Comment of Reviewer #1:

The same crystal standing on a lower hard plate gives rise to another boundary which however cannot support surface state. How to explain that with the bulk-boundary correspondence theorem?

Our response: We thank the reviewer for his/her instructive comments. The difference is coming from the nature of acoustic pseudospins. Unlike electrons with two intrinsic spins obeying $T^2=-1$ (T represents time-reversal operator) to satisfy the Kramers degeneracy, the acoustic pseudospins are constructed by spatial symmetries. It means that besides the bulk topology the surface condition also plays a key role to realize gapless surface Dirac states. In our case, the fields of acoustic pseudospins are mainly located at A surface layer (top surface in the main text), while B surface layer (bottom surface in the main text) does not support acoustic pseudospin states under hard boundary condition.

This is so-called “boundary-obstructed topological phase” or “fragile topological insulator”.

[Ref: E. Khalaf, W. Benalcazar, T. Hughes, and R. Queiroz, Boundary-obstructed topological phases, arXiv:1908.00011v1 (2019)];

[Ref: A. Alexandradinata, J. Holler, C. Wang, H. Cheng, and L. Lu, Crystallographic splitting theorem for band representations and fragile topological photonic crystals, arXiv:1908.08541v2 (2019)].

To further study the influence of the surface condition on gapless topological states, we numerically investigate various surface conditions as shown in Figs. R3-R5. In Fig. R3, decreasing the thickness gives rise to the trivial surface states (black lines), which will hybridize the nontrivial ones (red lines) to break the gapless property. This situation is similar to a 3D fragile topological photonic case [Fig. 8 of Ref: arXiv:1908.08541v2 (2019)]. Generally speaking, the gapless surface Dirac cones can always be constructed by choosing a proper surface condition to excite acoustic pseudospins in our model.

Fig. R3 | Truncated surface cases with the small spiral channel layer. **a**, Supercell configuration with hard boundaries. **b**, Projected bandstructures with various air layer thickness: $1.9l_0$, $0.5l_0$, $0.2l_0$, $0.1l_0$, $0.05l_0$, and $0.02l_0$. With decreasing the thickness, the trivial surface states (black lines) will hybridize the nontrivial ones (red lines) to break the gapless property. **c**, Supercell configuration with soft boundaries. **d**, Projected bandstructures with gapless surface states.

Fig. R4 | Truncated surface cases with the large spiral channel layer. **a**, Supercell configuration with hard boundaries. **b**, Projected bandstructures with various thickness: $1.9l_0$, $0.5l_0$, $0.2l_0$, $0.1l_0$, $0.05l_0$, and $0.02l_0$. With decreasing the thickness, the nontrivial surface states (red lines) may appear, however, the trivial ones (black lines) also exist. **c**, Supercell configuration with soft boundaries. **d**, Projected bandstructures with gapless surface states.

Fig. R5 | Truncated spiral channel cases. **a**, Supercell configuration and the projected bandstructures with truncated large spiral with hard boundary ($0.75s$). **b**, Soft boundary case ($0.25s$). **c**, Truncated small spiral case with hard boundary ($0.75s$). **d**, Soft boundary case ($0.25s$).

Action: To avoid misunderstanding, we have slightly revised the title to “Acoustic analogues of three-dimensional topological insulators”. We have also added a paragraph in the discussion section to address this issue. See:

Page 8

“It should be noticed that the pseudospins of our acoustic systems are based on spatial symmetries, which are unlike the electrons with two intrinsic spins naturally satisfying Kramer degeneracy⁴⁴. Thus, besides bulk topology, the surface condition also plays a key role to realize the gapless surface states.

In our case, the top surface of current model can support suitable acoustic pseudospin states while the bottom surface does not support these states for hard boundary condition, and vice versa for soft boundary condition⁴⁹. This is so-called boundary-obstructed topological phase⁴⁶ or fragile TI⁵⁰⁻⁵². On the other hand, although such fragile topological phase suffers from some limitations, it provides us an opportunity to develop on-demand gapless acoustic surface Dirac cones by choosing proper surface condition (Supplementary Materials Figs. S11-S18).”

Comment of Reviewer #1:

What’s the topological invariant of 2nd-order topological insulator?

Our response: We thank the reviewer for raising this important issue. Here, we use nested Wilson loops to show the high-order topology of the hinge states in our 3D acoustic system. Our topological acoustic hinge states base on the band folding mechanism of surface Dirac cones. So, the first two bulk bands below the bandgap of Fig. 1g in the main text are folded into six bands for the hinge state case.

For the trivial case (left panel of Fig. 4a in the main text), the unit cell (Fig. R6a) is three times larger than that of Fig. 1d in the main text. We numerically calculate the 2D Wannier bands of a Wilson loop along the z direction for these six bands shown in Fig. R6b. Three separated Wannier sectors (lower one, middle two and upper three bands) can be obtained, indicating the surface bandgap. To support gapless hinge states, at least two Wannier bands are needed in time-reversal invariant systems. Then, we calculate the nested Wannier loops for lower three and middle two bands along the y direction, respectively (Figs. R6c). We can see that these 1D nested Wannier bands show trivial character for hinge states with Wannier center located at $v_y=0$.

Fig. R6 | Nested Wilson loop for trivial case. a, Schematic of unit cell (left panel of Fig. 4a in the main text). **b,** Numerical 2D Wannier bands of a Wilson loop along the z direction. **c,** Numerical 1D Wannier bands of a nested Wilson loop for lower three bands (v_z^+ in **b**) and for middle two bands along the y direction.

But for the nontrivial case (right panel of Fig. 4a in the main text), two separated Wannier sectors (three bands in each one) can be obtained. We also calculate the nested Wannier loop for lower three bands along y direction as shown in Fig. R7. Interestingly, besides $v_y=0$, the Wannier centers can also be found at $v_y=\pm\pi$. It means that such hinge state is a phononic obstructed atomic limit case, which is different from trivial cases whose Wannier centers always locate at the origin. On the other hand, such 1D nested Wannier bands in our 3D acoustic system are similar to 1D Wannier bands of 2D fragile acoustic TI after considering a set of three bulk bands the below bandgap, which can be treated as a kind of acoustic analogue of 3D 2nd-order TI.

[Ref: M. Blanco de Paz, C. Devescovi, G. Giedke, *et. al.*, Tutorial: Computing topological invariants in

two-dimensional photonic crystals, arXiv:1912.00944v1 (2019)];

[Ref: H. Wang, G. Guo, and J.-H. Jiang, Band topology in classical waves: Wilson-loop approach to topological numbers and fragile topology, *New J. Phys.* **21**, 093029 (2009)];

[Ref: E. Khalaf, W. Benalcazar, T. Hughes, and R. Queiroz, Boundary-obstructed topological phases, arXiv:1908.00011v1 (2019)].

Fig. R7 | Nested Wilson loop for nontrivial case. **a**, Schematic of unit cell (right panel of Fig. 4a in the main text). **b**, Numerical 2D Wannier bands of a Wilson loop along z direction. **c**, Numerical 1D Wannier bands of a nested Wilson loop for lower three bands along y direction.

To further confirm the topology of the hinge states in our 3D acoustic system, we numerically calculate the Berry curvatures of the surface states near the center of surface Brillouin zone. For the trivial case (left panel of Fig. 4a in the main text), the Berry curvatures for two acoustic pseudospins calculated via two surface bands below the surface bandgap show its trivial character (Figs. R8a-b), with spin Chern numbers $C_{\pm}^s \approx 0$ (numerical results are ± 0.006 with a 24×24 mesh). But for the nontrivial case (right panel of Fig. 4a in the main text), the surface Berry curvatures show nontrivial (Figs. R8c-d), with spin Chern numbers $C_{\pm}^s \approx \pm 1$ (numerical results are ± 0.952 with a 24×24 mesh).

Fig. R8 | Berry curvatures of the surface states near the center of surface Brillouin zone **a-b**, Trivial case corresponding to the left panel of Fig. 4a in the main text. **c-d**, Nontrivial case corresponding to the right panel of Fig. 4a in the main text

Action: To avoid misunderstanding, we have revised the phrase using “acoustic analogue of 3D 2nd-order TI” to describe our system. We have also added these figures into supplementary materials.

Comment of Reviewer #2:

This paper studies a 3D acoustic structure that simulates a 3D topological insulator exhibiting surface Dirac cones and a second order topological insulator exhibiting 1D gapless hinge states. The design and experiment are both new. I particularly like the idea of folding two Weyl points to form a 3D Dirac point.

Our response: We do appreciate the reviewer for his/her positive comments on our work.

Comment of Reviewer #2:

1. The observed Dirac cone in Fig. 3, from my point of view, can probably arise from the honeycomb structure at the top surface (just like graphene), and thus may not necessarily link to the bulk topology.

Our response: We thank the reviewer for his/her instructive comments. Indeed, the honeycomb structure contributes to the surface states appearing at the top. However, it is not the sufficient condition for topological surface Dirac cones. On one hand, by directly stacking 2D topological honeycomb layers to 3D, it is hard to realize a 3D full bulk gap due to the z -direction coupling. On the other hand, additional symmetry breaking may realize a 3D full bulk gap, but it is not necessarily associated with topological surface Dirac cones. For example, if we use alternate chiral channels to obtain full bulk gap, only trivial topological states can be observed as shown in Fig. R9. In this case, S_6 symmetry is invariant while C_6 is broken to C_3 . However, in the main text, S_6 symmetry is broken while C_6 is kept.

Fig. R9 | Alternate chiral channel case. **a**, Alternate chiral channels used in the same layer to break the degeneracy. **b**, Bulk bandstructures with full gap. **c**, Supercell configuration. **d**, Projected bandstructures in the k_{xy} plane with hard and soft boundary condition, where no surface states can be observed. **e**, Supercell configuration with truncated surfaces. **f**, Projected bandstructures and Bloch field distributions with hard boundaries. **g**, Soft boundary condition. Only the trivial surface states can be observed in the cases **f** and **g**.

Here, we use bent Wilson-loop approach to study the bulk topology of our model with C_6 symmetry. Due to the C_6 symmetry, the maximal range of the eigenvalues of bent Wilson loop is $[-2\pi/3, +2\pi/3]$. As shown in Fig. R10, the eigenvalues of bent Wilson loop at $k_z=0$ and $k_z=\pi$ points equal to $\pm 2\pi/3$, showing that our model (z direction) belongs to weak topological phase.

[Ref: A. Alexandradinata and B. A. Bernevig, Berry-phase description of topological crystalline insulators, *Phys. Rev. B* **93**, 205104 (2016)].

Fig. R10 | Bent Wilson loop. **a**, Schematic of a bent loop in the 3D Brillouin zone of hexagonal lattice. **b**, Berry-phase spectrum of a bent Wilson loop along the z direction.

It should be noticed that the eigenvalues of bent Wilson loop in our model do not interpolate across the whole $[-2\pi/3, +2\pi/3]$ region. It means that the corresponding surface states cannot be gapless under arbitrary surface condition. This situation is similar to a 3D topological photonic case with C_{4v} symmetry. Although the bent Wilson loop is gapless [Fig. 7 of Ref: A. Alexandradinata, J. Holler, C. Wang, H. Cheng, and L. Lu, Crystallographic splitting theorem for band representations and fragile topological photonic crystals, arXiv:1908.08541v2 (2019)], their gapless surface states appearing at the top or bottom are dependent on the perfect-electric-conductor or perfect-magnetic-conductor boundary condition. [Fig. 5 of Ref: T. Ochiai, Gapless surface states originating from accidentally degenerate quadratic band touching, *Phys. Rev. A* **96**, 043842 (2017)].

Action: We added above figures and references into our supplementary materials (Figs. S19 and S25).

Comment of Reviewer #2: *Moreover, I don't understand why the bottom surface has no surface states. I hope the authors can clarify these points more clearly.*

Our response: We thank the reviewer for his/her comments. Our model is two-layer BA-stacking structure, where A represents the small spiral channel layer while B represents the large spiral channel layer. As shown in Fig. R11, the surface states prefer A surface layer under hard boundary condition, while they prefer B surface layer under soft boundary condition. The difference is coming from the nature of acoustic pseudospins. Unlike electrons with two intrinsic spins obeying $T^2=-1$ (T represents time-reversal operator) to satisfy the Kramers degeneracy, the acoustic pseudospins are constructed by spatial symmetries. It means that besides the bulk topology the surface condition also plays a key role to realize surface Dirac cone. In our case, the fields of acoustic pseudospins are mainly located at A surface layer (top surface in the main text), while B surface layer (bottom surface in the main text) does not support acoustic pseudospin states under hard boundary condition. This is so-called “boundary-obstructed topological phase” or “fragile topological insulator”.

[Ref: E. Khalaf, W. Benalcazar, T. Hughes, and R. Queiroz, Boundary-obstructed topological phases, arXiv:1908.00011v1 (2019)];

[Ref: A. Alexandradinata, J. Holler, C. Wang, H. Cheng, and L. Lu, Crystallographic splitting theorem for band representations and fragile topological photonic crystals, arXiv:1908.08541v2 (2019)].

Fig. R11 | Intact boundary cases with hard and soft boundaries. **a**, BA-stacking supercell configuration, where A represents the small spiral channel layer while B represents the large spiral channel layer. **b**, Projected bandstructures and Bloch field distributions with hard boundaries, where the surface states located at the top (A layer). **c**, Soft boundary case, where the surface states located at the bottom (B layer). **d**, AB-stacking supercell configuration. **e**, Hard boundary case, where the surface states located at the bottom (A layer). **f**, Soft boundary case, where the surface states located at the top (B layer).

Action: We added above figure and references into our supplementary materials (Fig. S15). To avoid misunderstanding, we have slightly revised the title to “Acoustic analogues of three-dimensional topological insulators”.

Comment of Reviewer #2:

2. In Fig. 2, I guess all surface states are excited in the measurement. So the real situation is more complicated than the simulation in Fig. 2f. For example, what is the spin-momentum locking in this situation?

Our response: We thank the reviewer for his/her comments. Indeed, the measured acoustic fields are more complicated than simulation results. As shown in Figs. R12-R13, we scan the acoustic fields in three different slices for straight and 120° bend propagation, respectively. These measurements agree well with the simulations except a little scattering in some directions. Generally, the backscattering is large suppressed, which confirms the acoustic pseudospin-momentum locking property. And, we can find the acoustic fields in slice 1 and 3 are weaker than those in slice 2 due to the point-like acoustic source used to excite the acoustic pseudospin.

Fig. R12 | Experimentally measured acoustic fields for straight propagation. **a**, Schematic of configuration to measure acoustic fields for three slices with $2h$, $3.5h$ and $5h$, respectively. **b**, Measured acoustic field of the three slices at various frequencies 5.8, 5.9 and 6 kHz, respectively.

Fig. R13 | Experimentally measured acoustic fields for 120° bend propagation. **a**, Schematic of configuration. **b**, Measured acoustic fields.

To further reveal the acoustic pseudospin-momentum locking in our model, we numerically analyze the pseudospin textures as shown in Fig. R14. The Acoustic pseudospins here are composed of in-phase and out-of-phase vibrations in two different layers, which can also be confirmed by recently published work about theoretical study of pseudospin phonons in 3D topological systems.

[Ref: Y. Liu, Y. Xu, and W. Duan, Three-dimensional topological states of phonons with tunable pseudospin physics, *Research* **2019**, 5173580 (2019).]

Fig. R14 | Acoustic pseudospin textures. **a**, Surface Dirac cone in the k_{yz} plane similar to Fig. 2c in the main text. **b**, Bloch field distributions for two slices (red and blue points in **a**). **c**, Acoustic pseudospin textures of the slice 1, which are composed of in-phase (e.g. points 1 and 5) and out-of-phase (e.g. points 3 and 7) vibrations in two different layers. **d**, Acoustic pseudospin textures of the slice 2. Here, the black arrows represent acoustic intensity vectors in the xy plane.

Action: We added above three figures in our supplementary materials (Figs. S3-S5).

Comment of Reviewer #3:

The work looks interesting. Nevertheless, clarification of some aspects could improve the presentation. In particular:

1. Abstract, lines 52-60: it would be helpful if the authors elaborate more on the differences between the current work and Refs. 29-32.

Our response: We thank the reviewer for his/her comments. Refs. 29-32 studied various acoustic Weyl systems, which can be treated as acoustic analogues of topological Weyl semimetals. Such topological semimetals are materials whose bulk bandstructures contain paired linearly touching points (Weyl points) with opposite Berry charges. But in our work, we focus on the 3D acoustic topological insulator system with bulk full band gap, where it can support gapless surface Dirac states in the gap.

Action: We have added one sentence to emphasize this issue. See:

Page 4

“Compared to 3D TIs with gapless surface Dirac cones in full bulk bandgaps, the bulk bandstructures of these Weyl systems contain paired linearly touching points (Weyl points) with opposite Berry charges.”

Comment of Reviewer #3:

2. Line 54: what do the authors mean by "whereas sound could be scattered into crystals due to the gapless bulk bandstructures that inevitably jeopardize the efficiency of sound transport."? The bulk bandstructures shown in Refs. 28-32 look gapped.

Our response: We thank the reviewer for his/her comments. Refs. 29-32 studied acoustic Weyl systems whose bulk bandstructures contain touching points at high symmetric points. In another words, there are no full bulk band gap in these systems. Although the directional band gap may exist with surface states along some directions, sound will be scattered into the bulk when meeting the defects or bends. Ref. 28 is hybrid case whose surface state along z direction is nonlinear. Therefore, the surface transmission along z direction shows small dip in the middle of the band gap [Fig. 3(d) of Ref. 28].

Action: We have added two sentences to explain this point. See:

Page 4

“Due to the lack of full bulk bandgaps, the sound could be scattered into crystals when meeting defects or bends, which inevitably jeopardize the efficiency of sound transport.”

Page 3

“...the topology of the extra out-of-plane direction is difficult to achieve the surface Dirac cone that is linear dispersion along an arbitrary direction^{27,28}.”

Comment of Reviewer #3:

3. Figure 1: subplot (a), the 3D Dirac point and the dispersion of the surface states looks quite similar to subplots (a) and (c) in Figure 1 in Ref. 28. For the surface states, what is different in current work in the ability to guide sound along the surface (which was impossible in Ref. 28)? Is the main novelty here are the hinge states?

Our response: We appreciate the reviewer for these comments. Indeed, they look somewhat similar visually. However, the underlying physics behind them is very different. In Ref. 28, the surface states are based on splitting of bulk semi Dirac cone. The corresponding surface state dispersion is linear in

the k_{xy} plane but quadratic along the k_z axis, which is not in the Dirac-like fashion. It means that the surface sound guiding along the z direction cannot be topological protected in principle. The surface dispersion can be written as

$$\omega \propto \sqrt{k_{x/y}^2 + k_z^4}.$$

But, in our current model, the surface dispersion is always linear along arbitrary direction, reading as

$$\omega \propto \sqrt{k_y^2 + k_{z/x}^2}.$$

To the best of our knowledge, this work is the first realization of acoustic topological insulators in 3D, which can exhibit 2D massless Dirac quasiparticles near the surface Dirac cone for a spinless system. The gapless acoustic hinge state is another important selling point of our work. Previous studies of high order acoustic topological states are all gapped, which cannot be used to guide sound without backscatterings.

Action: We have revised the color style of Fig. 1a in the main text to avoid misunderstanding.

Revised Fig. 1 | From Dirac cones to 3D 1st- and 2nd-order acoustic analogues of TIs. **a**, Left panel: visualization of a 3D Dirac cone for bulk bands, where the color scale represents frequency. Middle panel: 1st-order TI case by splitting the bulk Dirac degeneracy. Right panel: 2nd-order TI case after further splitting the surface degeneracy.

Comment of Reviewer #3:

4. Figures 2-3: it appears that the presented guided surface waves are a result of numerical simulations. Are there any measurements of such waves conducted in the experimental system?

Our response: We thank the reviewer for his/her comments. As shown in Figs. R15-R16, we scan the acoustic fields in three different slices for straight and 120° bend propagation, respectively. These measurements agree well with our simulations except a little scattering in some directions.

Fig. R15 | Experimentally measured acoustic fields for straight propagation. **a**, Schematic of configuration to measure

acoustic fields for three slices with $2h$, $3.5h$ and $5h$, respectively. **b**, Measured acoustic field of the three slices at various frequencies 5.8, 5.9 and 6 kHz, respectively.

Fig. R16 | Experimentally measured acoustic fields for 120° bend propagation. a, Schematic of configuration. **b**, Measured acoustic fields.

Regarding to the acoustic surface field distributions at the top surface of Fig. 3 in the main text, we scan the acoustic fields (the sample of Fig. S10) as shown in Fig. R17, which can also be used to obtain the Bloch momentum by a Fourier transformation (Fig. 3c). Dashed box in Fig. R16a matches well with the simulated result of Fig. 3e in the main text.

Fig. R17 | Experimentally measured acoustic fields for the top surface (the sample of Fig. S10). Measured acoustic fields at various frequencies **a**, 5.8 kHz, **b**, 5.9 kHz and **c**, 6 kHz.

Action: We added above three figures into our supplementary materials (Figs. S3, S4 and S7).

Comment of Reviewer #3:

5. Figure 4: it would be helpful if the authors explain more about the differences between (d)-top and (d)-bottom.

Our response: We thank the reviewer for his/her comments. The topological acoustic hinge states are based on the surface Brillouin zone folding of Fig.3 in the main text. Such supercell is constructed by two-layer BA-stacking structure, where A represents the small spiral channel layer while B represents the large spiral channel layer. As shown in Fig. R18, the surface states prefer A surface layer under hard boundary condition, while they prefer B surface layer under soft boundary condition. It means that the acoustic hinge states appear at the bottom under soft boundary condition with BA-stacking used in the main text (Fig. R19).

Fig. R18 | Intact boundary cases with hard and soft boundaries. **a**, BA-stacking supercell configuration, where A represents the small spiral channel layer while B represents the large spiral channel layer. **b**, Projected bandstructures and Bloch field distributions with hard boundaries, where the surface states located at the top (A layer). **c**, Soft boundary case, where the surface states located at the bottom (B layer). **d**, AB-stacking supercell configuration. **e**, Hard boundary case, where the surface states located at the bottom (A layer). **f**, Soft boundary case, where the surface states located at the top (B layer).

Fig. R19 | Soft boundary condition for hinge state compared to Fig. 4d in the main text. Simulated acoustic pressure distributions for the top and bottom hinges at a frequency of 5.81 kHz.

Action: We added above two figures into our supplementary materials (Figs. S15 and S23).

A list of our changes:**Text**

We have slightly revised the title to “Acoustic analogues of three-dimensional topological insulators”.

We have added a paragraph to address the bulk-boundary correspondence in our case in discussion section.

Several sentences have been added to emphasize the difference between our model and previous works in introduction section

Figures

According to reviewer’s suggestion, we have revised Fig. 1a in the main text.

References

Additional 7 references have been added in the main tet.

Supplementary information

Additional 15 figures have been added in supplementary materials.

REVIEWERS' COMMENTS:

Reviewer #1 (Remarks to the Author):

It is a clever way to implement the band-folding into a Weyl system for building up an acoustic analogue of three-dimensional topological insulator. Their calculated bulk topology is consistent with observed surface/hinge states. I have no more comments and would suggest to publish as it is.

Reviewer #2 (Remarks to the Author):

I think the authors have answered my questions. I have no further comments.

Reviewers' comments:

Comment of Reviewer #1 (Remarks to the Author):

It is a clever way to implement the band-folding into a Weyl system for building up an acoustic analogue of three-dimensional topological insulator. Their calculated bulk topology is consistent with observed surface/hinge states. I have no more comments and would suggest to publish as it is.

Our response: We do appreciate the reviewer for his/her support.

Reviewer #2 (Remarks to the Author):

I think the authors have answered my questions. I have no further comments.

Our response: We thank the reviewer for his/her efforts in reviewing our manuscript.